# The Pathogenesis, Molecular Mechanisms, and Therapeutic Potential of the Interferon Pathway in Systemic Lupus Erythematosus and Other Autoimmune Diseases

**DOI:** 10.3390/ijms222011286

**Published:** 2021-10-19

**Authors:** Madhu Ramaswamy, Raj Tummala, Katie Streicher, Andre Nogueira da Costa, Philip Z. Brohawn

**Affiliations:** 1Translational Science and Experimental Medicine, BioPharmaceuticals R&D, AstraZeneca, Gaithersburg, MD 20878, USA; andre.dacosta@astrazeneca.com (A.N.d.C.); zach.brohawn@astrazeneca.com (P.Z.B.); 2Respiratory, Inflammation & Autoimmunity, BioPharmaceuticals R&D, AstraZeneca, Gaithersburg, MD 20878, USA; Raj.Tummala@astrazeneca.com; 3Translational Medicine, BioPharmaceuticals R&D, AstraZeneca, Gaithersburg, MD 20878, USA; katie.streicher@astrazeneca.com

**Keywords:** interferon, systemic lupus erythematosus, autoimmunity

## Abstract

Therapeutic success in treating patients with systemic lupus erythematosus (SLE) is limited by the multivariate disease etiology, multi-organ presentation, systemic involvement, and complex immunopathogenesis. Agents targeting B-cell differentiation and survival are not efficacious for all patients, indicating a need to target other inflammatory mediators. One such target is the type I interferon pathway. Type I interferons upregulate interferon gene signatures and mediate critical antiviral responses. Dysregulated type I interferon signaling is detectable in many patients with SLE and other autoimmune diseases, and the extent of this dysregulation is associated with disease severity, making type I interferons therapeutically tangible targets. The recent approval of the type I interferon-blocking antibody, anifrolumab, by the US Food and Drug Administration for the treatment of patients with SLE demonstrates the value of targeting this pathway. Nevertheless, the interferon pathway has pleiotropic biology, with multiple cellular targets and signaling components that are incompletely understood. Deconvoluting the complexity of the type I interferon pathway and its intersection with lupus disease pathology will be valuable for further development of targeted SLE therapeutics. This review summarizes the immune mediators of the interferon pathway, its association with disease pathogenesis, and therapeutic modalities targeting the dysregulated interferon pathway.

## 1. The Interferon Pathway and Its Role in Autoimmunity: Systemic Lupus Erythematosus and Beyond

### 1.1. Interferons Are a Complex Family of Cytokines with Antiviral and Immunomodulatory Functions

Interferons are cytokines with pleiotropic roles in immune regulation that can be categorized into three families (type I, II, or III) based on sequence homology [1]. There is one member of the type II interferon family (interferon [IFN]-γ) and four members of the type III family (IFN-λ1–4). The type I interferon family is the largest, comprising five classes (IFN-α, IFN-β, IFN-ω, IFN-κ, and IFN-ε), of which there are 12 further subgroups of IFN-α proteins in humans [2].

Type I interferons mediate critical antiviral responses and play an immunomodulatory “bridging” role that links both the innate and adaptive immune systems [2]. The innate immune response to viral infection transiently induces high levels of type I interferon expression [3,4]. In brief, the type I interferon response is activated when nucleic acids (RNA, DNA) derived from viruses trigger intracellular pathways mediated by pattern recognition receptors (including Toll-like receptors [TLRs]) to converge on the expression of a type I interferon [3,4] (Figure 1).

The type I interferon then binds to type I interferon α/β receptors (IFNAR) on effector immune cells of the innate and adaptive systems [2,5,6]. The interferon-IFNAR1 complex activates an intracellular signaling cascade, mediated in part by janus kinase (JAK) and signal transducer and activator of transcription (STAT) phosphorylation cascades, which trigger expression/modulation of interferon-regulated antiviral genes [2]. These interferon-stimulated gene products activate the antiviral innate and adaptive immune systems and trigger antigen-specific antiviral T-cell and B-cell responses [7,8,9].

### 1.2. Dysregulated Type I Interferon Signaling Is Associated with Loss of Immune Tolerance and Autoimmunity

While the transient induction of type I interferon genes in response to viral stimuli is key for host immunity, chronic or dysregulated activation of the type I interferon pathway can contribute to the development of systemic lupus erythematosus (SLE) [2,10,11,12]. In patients with SLE and other autoimmune diseases, this aberrant type I interferon production is triggered by self-nucleic acids in autoimmune complexes, which are often generated as a by-product of a deficiency in apoptotic cell clearance or from dysregulated neutrophil functions [13,14,15,16]. Combined with a loss of the negative feedback signal usually required to “switch off” type I interferon signaling, as well as variants in interferon-related genes [2,17], this chronic type I interferon production stimulates the innate and adaptive immune systems in response to a self-derived trigger, enhancing the original stimulus and contributing to the loss of immune tolerance that is characteristic of autoimmunity [7,17]. This process induces the production of further autoantibodies [18] and type I interferon in a positive autoamplification loop, or vicious cycle [7,14,19].

A build-up of autoimmune complexes can induce systemic and localized inflammation. In patients with SLE, deposits of autoimmune complexes and subsequent inflammation in the tissues contribute to irreversible organ damage and substantial disease burden [20] (Figure 2). Patients with SLE can have a heterogeneous array of organ manifestations, with the most commonly affected organs being the skin, joints, and kidneys [21,22,23]. Of note, non-European patients with SLE often have increased autoantibody reactivity compared with European patients, which may explain why patients who are Black, Asian, or Hispanic present with more severe lupus phenotypes, a greater number of organ manifestations, and a more rapid accumulation of organ damage than patients who are White [24].

While the cellular source of chronic type I interferon production in SLE and other autoimmune diseases is incompletely understood, the predominant source in SLE is likely the plasmacytoid dendritic cell (pDC) [19,25]. pDCs play a predominant role in the aforementioned autoamplification loop that drives type I interferon production and downstream autoantibody generation, as antigen-presentation and type I interferon production take place simultaneously in pDCs and can recruit new pDCs to tissues [19]. Other cell types are also known to contribute, including non-hematopoietic cells, which are thought to play a key role early in disease pathogenesis before clinical autoimmunity [26].

### 1.3. Multiple Lines of Evidence Support the Role of Interferons in SLE and Autoimmunity

Over the past 40 years, progressive research has contributed to our understanding of the role of interferons in the immunopathogenesis of SLE and other autoimmune diseases (summarized in Figure 3).

#### 1.3.1. Elevated Interferon Protein Is Detected in Patients with Autoimmune Diseases

A role for type I interferons in autoimmunity was first proposed in 1979, when Hooks et al. identified elevated levels of interferon in the serum of patients with SLE, rheumatoid arthritis, systemic sclerosis (SSc), and Sjögren’s syndrome [27]. In 1982, the serum interferon in patients with SLE was identified as IFN-α [28]. More recently, type I interferon protein or transcript signatures have been identified in patients with SLE, myositis (particularly dermatomyositis (DM)), lupus nephritis (LN), SSc, and Sjogren’s syndrome [29,30,31,32,33,34], and low levels of type III interferons have also been identified in patients with SLE [35].

Studies have identified variation in interferon protein expression across SLE racial groups, with greater IFN-α expression in patients with non-European versus European ancestry [36,37]. Furthermore, the autoantibody profile differs across ancestral groups, suggesting alternative mechanisms of interferon induction across different racial groups of patients with SLE [37].

#### 1.3.2. Therapeutic Interferon Can Induce/Exacerbate Autoimmunity

Following initial observations in the 1970s of elevated interferon proteins in patients with autoimmune disease, the next line of evidence for a role of interferons came from observational studies and case reports beginning in 1985. These studies described individuals who developed autoantibodies and autoimmunity (including SLE and SSc) following treatment with IFN-α for malignancies or hepatitis [38,39,40,41]. A randomized, placebo-controlled trial revealed that treatment with IFN-α exacerbated skin and lung manifestations compared with standard therapy in patients with SSc, further supporting a role for type I interferon in disease pathogenesis [42]. As well as inducing the development of SLE and SSc, IFN-α treatment has been rarely associated with the induction of myositis, an autoimmune disease causing muscle weakness [38,43]. A case study described a patient developing inclusion body myositis following treatment with IFN-α, with symptoms exacerbated after repeat dosing [43]. Furthermore, electron microscopy following intramuscular IFN-α administration identified the formation of “lupus inclusions”, a common manifestation of DM, in muscle endothelium [38,43,44].

#### 1.3.3. Interferon Gene Signatures Are Upregulated in Patients with Autoimmune Diseases

While measurement of interferon chemokines and reports of unwanted side effects of IFN-α therapy were fundamental in establishing a role for type I interferons in patients with SLE, there are practical challenges associated with quantifying IFN-α protein, which is often present at very low concentrations. These challenges necessitated the development of an interferon gene signature (IFNGS) that could be accurately and accessibly measured as an alternative output for type I interferon pathway activity [45]. In the past 20 years, dysregulated IFNGS expression in blood and affected tissues has been detected in 52–87% of adult patients with SLE [30,45,46,47,48,49,50,51,52,53]. Studies have characterized the key IFN-α/β-inducible transcripts that are overexpressed in whole blood or tissue of patients with SLE compared with healthy controls (Figure 4) [54,55], the dysregulation of which may contribute to SLE immunopathogenesis by way of mechanisms yet to be fully understood [10].

The proportion of patients with SLE who have an elevated IFNGS expression varies with age at diagnosis, disease manifestations, and race; for example, more than 90% of pediatric patients with SLE [56] and 67–83% of adult patients with renal manifestations (LN) [47,57] have an elevated IFNGS expression. An elevated IFNGS expression is also more prevalent among Black and Asian patients than White patients with SLE [58].

Dysregulated type I interferon signaling is also extensively reported for autoimmune diseases other than SLE [59], with an elevated IFNGS expression broadly detected in 68% of patients with SSc, 66% of patients with DM, 61% of patients with polymyositis (PM), and 33% of patients with rheumatoid arthritis, as shown in Figure 5 [30]. Further support for the centrality of interferon axis dysregulation in these autoimmune diseases can be gathered from the finding that type I IFNGS expression could be identified from the earliest SSc disease phases in one patient cohort, even before the emergence of characteristic markers of SSc skin fibrosis that signify the start of organ damage [33]. In the same study, the presence of the IFNGS in monocytes correlated with markers of disease progression, leading the authors to conclude that the type I interferon pathway contributes to both the pathogenesis and progression of SSc [33]. Similarly, in two independent studies, an elevated IFNGS expression was associated with subsequent SLE diagnosis among individuals deemed at high risk of developing an autoimmune condition, supporting a role for interferons in early SLE pathogenesis and progression [60,61].

An elevated type I IFNGS expression in patients with SLE is reported more frequently than an elevated type II or III IFNGS. However, the downstream effects of type I, II, and III interferon signaling overlap considerably, with a number of common genes activated by all three interferon families [62,63,64]. As such, the type I IFNGS expression observed in patients with SLE may also reflect type II and III interferon activity, and assessment of IFN cytokine levels in future studies could help to pinpoint the precise contributing interferon mediators.

#### 1.3.4. Many Immune Cell Types Express Interferon Gene Signatures in Autoimmune Disease

Recently, single-cell analyses have been utilized to investigate the cellular source of the elevated type I interferon gene signature detected in serum and tissue of patients with SLE or LN (i.e., the cells responding to the aberrantly produced pDC-derived type I interferon). These studies identified elevated interferon-regulated gene expression in a broad range of immune cell types (e.g., B cells, natural killer cells, monocytes, T cells, conventional dendritic cells, pDCs, neutrophils, and low-density granulocytes) [65,66,67,68]. Furthermore, single-cell RNA sequencing of renal biopsies from patients with LN identified type I interferon responses in keratinocytes and tubular cells that distinguished patients with LN from healthy controls [69]. The broad range of cell types expressing the IFNGS likely reflects the ubiquity of IFNAR1 expression across many immune cell types [2,6]. Signal blockade via the IFNAR1 thereby has the potential to affect disease biology across a range of immune cell types, including B cells, T cells, pDCs, and natural killer cells [6]; these studies also highlight the potential for targeting specific IFN-producing cell types (such as pDCs, discussed in a later section).

#### 1.3.5. Polymorphisms in Interferon-Related Genes Associate with Increased Risk of Autoimmune Disease

Within the past 13 years, genome-wide association and pathway-centered sequencing studies have identified many polymorphisms in interferon pathway genes that predispose an individual to developing SLE, including polymorphisms affecting the *STAT4*, *IRF3*, and *IFNA1* genes [70,71,72,73,74]. While single polymorphisms may not be sufficient to cause autoimmunity alone, accumulating numbers of risk variants may cause the immune system to become overwhelmed, leading to immune dysregulation [24,75]. Furthermore, interferon signaling upregulation inpatients with LN may trigger nephropathy associated with apolipoprotein L1 (*APOL1*) risk-alleles (which are prevalent in individuals with Sub-Saharan African ancestry) by upregulating *APOL1* mRNA expression and increasing genotype-associated disease severity [76]. Risk-associated variants in interferon-related genes have also been identified for other autoimmune diseases, including SSc, for which family history remains the strongest known risk factor [32].

#### 1.3.6. Type I Interferon Gene Signature Expression Occurs within a Broader Context of Immune Dysregulation

Decades of profiling gene and protein expression have confirmed the molecular heterogeneity of SLE, and recent studies have attempted to contextualize interferon upregulation within broader immune dysregulation. In a transcriptomics study of pediatric patients with SLE, “personalized immunomonitoring” over time revealed gene expression modules, supported by genotypes, that enabled patient stratification by disease activity. These modules included interferon response genes as well as genes categorized by plasmablast, cell cycle, neutrophil, histone, and B-cell functions [77]. A separate investigation of longitudinal gene expression identified multiple gene modules associated with neutrophil and lymphocyte activity, and differentially associated with type I interferon activity, that allowed adult and pediatric patients with SLE to be stratified by disease activity [78]. Another study assessing gene co-expression across different immune cell types in patients with SLE identified coordinated interferon responses and cross-cell type correlations linking T-helper cell markers and B-cell responses that, in turn, correlated with disease severity [79]. These findings are relevant because stratification by disease activity may provide rationale for clinical trial failures/successes and may help to identify subpopulations suited for targeted therapies.

### 1.4. Interferon Dysregulation Associates with Clinical Disease Activity and Response to Standard Therapy in Patients with SLE

A wealth of literature supports an association between the presence/extent of type I interferon signaling dysregulation and lupus disease activity. Multiple studies assessing independent cohorts of patients with SLE identified higher global disease activity scores for patients with an elevated IFNGS expression/activity, compared with patients who had a normal IFNGS expression [30,45,47,65]. Greater disease activity was also identified in patients with SLE who had type I interferon protein signatures [29,31], and IFN-λ1 protein levels also correlated with global disease activity in patients with LN and SLE [80]. 

Interferon pathway dysregulation also associates with the presence of organ-specific disease manifestations in patients with SLE and LN, including kidney, skin, and cardiovascular disease (reviewed in further detail in [2]). pDC infiltrates were detected in the affected kidneys of patients with LN [81], and multiple studies identified an association between serum or kidney-specific type I IFNGS expression and the presence and extent of LN disease activity [66,80,82]. In addition to type I interferons, the detection of IFN-γ-producing autoreactive T cells in the urine of patients with LN also suggests infiltration in inflamed kidneys [83]. The IFNGS expression in blood also correlated with skin disease activity in patients with subacute cutaneous lupus erythematosus (CLE) [84], and multiple studies identified an elevated IFNGS expression in the affected skin of patients with SLE, using both traditional biopsy approaches and, more recently, using non-invasive tape strip sampling [30,85,86]. Notably, IFN-κ production by keratinocytes may be an early step in driving lupus skin disease, where IFN- κ is proposed to activate dendritic cells, amplify IFN-α responses, and contribute to skin photosensitivity [26,87]. Type I interferon has also been implicated in cardiovascular disease in patients with SLE, with multiple studies identifying an association between type I interferon dysregulation and subclinical and clinical cardiovascular disease markers, independent of known cardiovascular risk factors [88,89,90].

While the role of type I interferon in kidney, skin, and cardiovascular disease is established, the immunopathogenesis of lupus joint manifestations is likely more complex, as multiple studies have detected no significant correlation between type I interferon pathway activity (assessed using serum IFN-α activity or interferon gene module expression) and SLE joint involvement using independent patient cohorts [36,48]. Furthermore, local IFN-β production in joints may even have a protective effect in patients with rheumatoid arthritis [91]. However, joint improvement in patients with SLE who received a therapeutic antibody that targeted the type I interferon pathway (discussed later) supports a role for type I interferon in lupus arthritis [50,51,52].

SLE is characterized by unpredictable longitudinal fluctuations in organ-specific and global disease activity, with periods of high disease activity known as flares [92,93]. While the relationship between type I interferon signaling and lupus flares is complex, studies have identified associations between type I interferon activity and longitudinal disease activity changes. For example, high levels of the IFN-α protein were associated with symptom relapse in one cohort of patients with SLE [94], while an elevated IFNGS expression predicted SLE flares in a separate cohort [95]. Despite the sporadic increases in disease activity observed in patients with SLE, the IFNGS expression tends to be stable over time in most patients receiving standard therapies. For example, in phase two and three trials of the anti-IFNAR1 antibody, anifrolumab, the IFNGS expression remained stable throughout the treatment period among patients in the control group who received background standard therapy with oral glucocorticoids and/or immunosuppressants (Figure 6) [50,51,52].

Although an IFNGS generally remains stable over time in patients receiving standard therapies, the extent of interferon dysregulation can affect the standard therapy dosage and response. Pooled data from two randomized controlled trials showed that more patients with SLE who had an elevated IFNGS expression were taking high doses of glucocorticoids (≥10 mg/day) than patients with low levels of IFNGS expression [96]. In a separate single-cell transcriptomic analysis of renal tissue from patients with LN, the patients who did not respond to standard therapy had a greater expression of type I interferon response genes in tubular cells than those who achieved at least a partial response [69]. Similarly, levels of IFN-α but not levels of IFN-λ1 protein correlated with poor response to immunosuppressive therapy in patients with SLE [80], while a reduction in serum IFN-γ protein levels was associated with response to ustekinumab, an anti-interleukin-12/-23 monoclonal antibody (mAb), in a phase two SLE trial [97]. 

Together, multiple lines of evidence support the fundamental role of interferons in autoimmunity and mediating response to standard therapies and highlight the therapeutic potential of targeting the type I interferon pathway in autoimmune diseases. 

## 2. Therapeutic Highlights: Past, Present, and Future

### 2.1. Current Standard Therapies, Approved Biologics, and Impact on Interferon Signaling

Currently, the approved standard therapies for patients with SLE include glucocorticoids, antimalarials, and immunosuppressants, for which usage recommendations are based on expert opinion and the literature supporting their effectiveness in treating these patients [92,93]. Some evidence shows that antimalarial and immunosuppressant agents modulate interferon pathway activity as part of their broad mechanism of action, whereas data assessing the effect of glucocorticoid use on interferons are limited (summarized in Table 1).

The largest body of evidence available for the modulation of type I interferons by standard therapies is for the antimalarial agent hydroxychloroquine (HCQ), which prevents the maturation of endosomes and disrupts the activity of endosome-resident TLRs, key components in type I interferon production [25]. Multiple studies have reported the inhibition of IFN-α production or IFNGS expression in patients with SLE or Sjögren’s syndrome following HCQ treatment [25,99,100]. However, the extent to which interferon inhibition by HCQ relates to its clinical effectiveness remains to be investigated.

The immunosuppressant drug mycophenolate mofetil (MMF) was also shown in one recent study to non-specifically inhibit the production of IFN-α by pDCs in a dose-dependent manner, and this was confirmed by transcript levels [98]. However, this study assessed pDCs derived from healthy volunteers only; therefore, future research is required to investigate the effect in an autoimmune environment (where patients may have dysfunctional pDCs [120]), and to investigate the downstream effect of IFNGS expression on MMF effectiveness.

Although use of standard therapies is associated with short-term benefits and these therapies are recommended by current treatment guidelines, not all patients will respond to them, and the use of these agents is often accompanied by adverse effects [92,93]. Glucocorticoid use, in particular, is associated with a significant burden, as high dosages (prednisone ≥ 7.5 mg/day or equivalent) are associated with an increased risk of organ damage compared with lower dosages [121,122,123].

Due to the unpredictable response and safety issues with standard therapies, there remains a need for additional therapeutic options to treat patients with SLE. Targeted therapeutics and precision medicine may be particularly valuable for a disease such as SLE, for which disease course, therapeutic response, and underlying immune mechanisms are both complex and heterogeneous, suggesting that a “one size fits all” approach may not be appropriate [124]. The need for precision medicine is further illustrated by the low response rates to standard therapies in patients with SLE [125].

Approval of belimumab, the first targeted biologic approved for use in patients with active SLE or LN despite standard therapy, represented a therapeutic milestone [126,127]. Belimumab is a mAb that inhibits the soluble counterpart of the B-lymphocyte stimulator (BLyS), which is a key mediator of B-cell differentiation and survival, and thus dampens B-cell mediated autoimmunity [126,127]. A recent post hoc analysis of phase three trial data indicated that baseline BLyS and IFN-1 mRNA levels were highly correlated and that the Systemic Lupus Erythematosus Responder Index ≥ 4 (SRI(4)) response to belimumab for IFNGS-high patients was consistent with the response overall [128], suggesting an indirect interaction between BLyS inhibition and interferon modulation.

### 2.2. An Interferon-Targeting Therapeutic Is Now Approved for the Treatment of SLE and Others Are in Clinical Development

The role of type I interferons in autoimmunity highlights the therapeutic potential of targeting the type I interferon pathway in autoimmune diseases. While standard therapies and belimumab may have indirect effects on type I interferon signaling as part of their broad mechanism of action, a number of targeted therapeutics that inhibit type I interferon signaling, either directly or indirectly, are at various stages of clinical development (summarized in Table 1). A timeline showing the clinical development of interferon-targeted biologics is presented in Figure 7. These molecules include biologics that directly target IFN-α chemokine or the IFNAR, and biologics that block the production of type I interferon by inhibiting or depleting the pDC function. Small molecule inhibitors, specifically JAK inhibitors, also indirectly target intracellular mediators of the type I interferon/IFNAR1 axis.

As of 2021, four direct type I interferon-targeting therapies have been developed for the treatment of patients with SLE, myositis, or LN, and have demonstrated variable clinical benefits. IFN-α kinoid (IFN-K) is an immunotherapeutic vaccine that induces the production of polyclonal anti-IFN-α antibodies to subsequently block type I interferon activity and normalize the IFNGS [107,108]. IFN-K was recently assessed in a phase two trial of patients with active SLE despite standard therapy and a positive IFNGS. Although more patients attained a lupus low disease activity state (LLDAS) or glucocorticoid tapering with IFN-K than with the placebo, the trial did not meet its clinical primary endpoint (modified British Isles Lupus Assessment Group (BILAG)-based Composite Lupus Assessment (BICLA) response) [109]. The development of IFN-K has since been discontinued and no further trials are planned.

Rontalizumab is a humanized mAb and sifalimumab is a fully human mAb that inhibit IFN-α and suppress IFNGS expression [53,110,111,129]. Rontalizumab treatment was associated with moderate clinical benefit in a phase two trial of patients with SLE with low IFNGS scores [111]; however, neither primary nor secondary endpoints (BILAG-2004 reduction, SLE Responder Index ≥ 4 (SRI(4)) response at Week 24) were met for all patients or for IFNGS-high patients [111], and development was discontinued. The study design may have affected the outcomes. Patients received the study drug intravenously for 24 weeks during stage one, followed by re-randomization and subcutaneous administration of study drug for a further 24 weeks in stage two [111,130]. The change from intravenous to subcutaneous administration halfway through the trial may have led to inadequate dosing, which could explain the lack of response for IFNGS-high patients [111,130]. Additionally, all patients taking immunosuppressive agents at baseline were required per protocol to discontinue, which could have confounded efficacy estimates, especially accounting for the 24-week duration of treatment prior to efficacy assessment, which is shorter than most other SLE trials [130]. This, combined with a small sample size, may have contributed to the failed primary outcome [111,130].

More promising results were observed with sifalimumab, which was associated with clinical benefit for patients with SLE and those with myositis [53,129]. In a phase 2b trial of patients with moderate to severe SLE, sifalimumab suppressed the IFNGS, and primary and secondary endpoints were met, with a greater proportion of patients achieving an SRI(4) response at Week 52 with sifalimumab versus placebo, accompanied by other improvements in skin disease activity and a reduction in swollen and tender joint counts [53]. In a phase one trial of patients with DM and PM, sifalimumab suppressed the IFNGS, and the extent of IFNGS suppression in muscle and blood was associated with an improvement in manual muscle test scores [129]. Despite the positive phase two results with sifalimumab, further development was paused in favor of the more substantial IFNGS neutralization observed with the anti-IFNAR1 antibody, anifrolumab.

In contrast to sifalimumab and rontalizumab, which target IFN-α, anifrolumab, the first US Food and Drug Administration (FDA)- and Japanese Ministry of Health, Labour and Welfare (MHLW)-approved therapeutic targeting the type I interferon pathway, is a fully human IgG1κ mAb that blocks all type I interferon/IFNAR1 signaling [103,104,105,106]. Anifrolumab rapidly internalizes the IFNAR1 and sterically inhibits the IFNAR1, preventing interferon/IFNAR1 complex formation, and ultimately suppressing the IFNGS [103,104]. Anifrolumab disrupts the type I interferon auto-amplification loop that can trigger the loss of immune tolerance and autoimmunity, as it blocks type I interferon signaling in pDCs and suppresses the induction of type I interferons following immune complex stimulation [103]. In addition, anifrolumab partially inhibits upregulation of costimulatory molecules and the production of pro-inflammatory cytokines by pDCs [103]. As well as inhibiting the type I IFN-pDC axis of the innate immune system, anifrolumab suppresses B-cell survival factors (e.g., BLyS, B lymphocyte chemoattractant (BLC)), inhibits B-cell differentiation, and suppresses T-cell activation, all of which are adaptive immune mediators of autoimmune pathogenesis [103,131,132].

Anifrolumab is indicated for the treatment of adult patients with moderate to severe SLE who are receiving standard therapy [104]. In phase two and phase three trials, anifrolumab treatment was associated with clinical benefit across multiple efficacy endpoints for patients with moderate to severe SLE despite standard therapy, consistently showing a greater improvement in overall disease activity as assessed by the BICLA response at Week 52 compared with a placebo [50,51,52]. While the BICLA response rates with anifrolumab were consistently higher than with a placebo, regardless of the extent of pharmacodynamic IFNGS expression, a positive relationship between greater IFNGS suppression and the magnitude of BICLA response rates was apparent [133]. Improvements were also seen for skin and joint disease activity with anifrolumab, and more patients were able to taper oral glucocorticoids to a low dosage threshold of ≤7.5 mg/day with anifrolumab than a placebo [50,51,52].

As well as showing improvements in clinical outcomes, anifrolumab treatment was associated with reduced levels of inflammatory and cardiometabolic disease markers in a phase two trial [90,131]. Patients with SLE were at a three- to four-fold greater risk of experiencing cardiovascular events than healthy controls (adjusting for known risk factors such as age, sex, and smoking history), as a result of immune-mediated vascular damage and premature atherosclerosis [134]. Anifrolumab treatment significantly modulated levels of multiple subclinical markers of cardiometabolic disease that contribute to SLE vasculopathy, suggesting that inhibiting type I IFN signaling could reduce atherosclerosis and cardiovascular markers in patients with SLE [90].

As well as a direct blockade of interferon/IFNAR1, strategies to inhibit type I interferon production from pDCs are in development for the treatment of patients with SLE and CLE. VIB7734 is a pDC-depleting mAb engineered for enhanced effector function that targets the pDC-specific marker immunoglobulin-like transcript 7 (ILT7) [112]. In a phase one trial that enrolled patients with CLE, VIB7734 depleted levels of circulating and tissue-resident pDCs, decreased local type I interferon production, and improved clinical disease activity [112].

Another mAb under development, BIIB059, binds the pDC-specific blood dendritic cell antigen 2 (BDCA2) receptor [114,115]. Unlike VIB7734, BIIB059 induces rapid BDCA2 internalization and inhibits type I interferon production without depleting pDCs [114]. In a phase one trial that enrolled patients with SLE, BIIB059 decreased the IFNGS expression in blood and normalized levels of interferon response proteins in the affected skin tissue [114]. In a phase two trial, BIIB059 treatment was associated with a reduction in the number of active joints and higher SRI(4) response rates compared with a placebo in patients with SLE [116]. In the same trial, BIIB059 treatment reduced skin disease activity scores compared with a placebo in patients with CLE [135]. These positive phase two results support continued clinical development of BIIB059 in patients with SLE and CLE [116,135].

In an alternative approach to targeting interferons, multiple JAK inhibitors (JAKi), which act downstream in the interferon signaling pathway, are in development. In a phase two trial of baricitinib (a JAK1- and JAK2-selective inhibitor), gene expression profiling revealed the suppression of interferon-responsive genes following 24 weeks of study treatment, in addition to the suppression of JAK-STAT signaling and other genes involved in SLE immunopathogenesis [119]. Similarly, the treatment of patients with SLE with tofacitinib (which inhibits JAK3, as well as JAK1 and JAK2 to a lesser extent) decreased the IFNGS expression in a phase one trial [117], and multiple JAKis were shown to suppress the IFNGS in a study of off-label use in pediatric patients with SLE [118].

Although the aforementioned interferon-targeting therapies are in clinical development for the treatment of patients with autoimmune disease, an interferon blockade also has the potential for clinical benefit in indications beyond autoimmunity. Recently, sustained type I interferon signaling was shown to cause resistance to immunotherapies in patients with melanoma [136]. Further investigations are needed to establish whether an interferon blockade could be beneficial in patients with cancer who have interferon-mediated therapeutic resistance.

### 2.3. The Future of Interferon Blockade and Lupus Therapies

The range of different interferon-targeting molecules in clinical development is perhaps due to the unprecedented clinical trial activity in SLE and other autoimmune diseases over the past two decades, in which trials are targeting varied mechanisms with biologics, kinase inhibitors, and other small molecule compounds [137]. While drug discovery in SLE has historically lagged behind other diseases, predominantly owing to challenges associated with measuring lupus disease activity in a heterogeneous disease, the vast number of new trials and improvements in trial outcome measures has led to a series of successful lupus trials with anifrolumab, voclosporin, and belimumab [137,138]. These recent successes, together with the sheer volume of ongoing clinical trials, increases the chance of additional drugs becoming available to patients within the next decade, illustrated by the recent approvals of belimumab and voclosporin for the treatment of patients with LN [137].

Future drug approvals may include interferon-targeting agents, based on the numerous interferon-targeting molecules (direct or indirect) currently in development and the recent successes seen in both early- and late-phase trials [50,112,135]. However, not all patients respond to type I IFN-targeting agents, and there is a need to understand the role played by other pathways and how these interact with type I IFN signaling.

Broader changes in lupus standard therapy use will likely emerge as more targeted therapeutics become available. Consistent with current trends, glucocorticoid use will likely decrease further in the next decade among patients with SLE, based on the recommendations of updated treatment guidelines [92,93]. Although glucocorticoid use is common among patients with SLE with an elevated IFNGS expression [96], type I interferon inhibition with anifrolumab reduced the need for glucocorticoid use in patients with SLE [50,51,52], highlighting the glucocorticoid-sparing capability of targeted anti-IFN therapies for some patients.

As we continue to deepen our understanding of lupus immunopathogenesis, stratification by immunophenotype will become increasingly commonplace in future SLE trials. In phase two and three anifrolumab trials, 75–83% of patients were IFNGS-high at screening, reflecting the natural distribution in this patient population [50,51,52]. A direct comparison of the anifrolumab treatment response for IFNGS-high versus IFNGS-low patients was limited by the small sample size of the IFNGS-low subgroup in the phase three anifrolumab trials. However, a post hoc analysis of pooled data from two phase three trials indicated that the magnitude of the treatment difference for anifrolumab versus placebo was greater among the IFNGS-high patients than the IFNGS-low patients, mostly attributable to the low placebo response rates with standard therapy in the IFNGS-high patients [50,51,52]. The availability of additional targeted therapeutic options will represent a further milestone for patients with dysregulated type I interferon signaling, who tend to have severe disease and suboptimal responses to standard therapy [30,45,69,80,96]. As an increased prevalence of type I interferon dysregulation has been identified in patients who are non-White versus White and treatment response can vary by race [24], future trials of targeted therapies should also monitor treatment response across racial groups.

## 3. Conclusions

The type I interferon pathway plays a major role in the etiology of SLE and many other autoimmune diseases. Patients with SLE with dysregulated type I interferon activity often have more active disease, which is resistant to standard therapies, compared with patients without dysregulated interferon signaling [30,45,47,65,80]. Over the past 40 years, extensive research from the lupus community has made substantial progress in deconvoluting the immune complexity of SLE and identifying key molecular and cellular components of type I interferon signaling. These efforts, combined with the unprecedented SLE clinical trial activity in the past two decades, have culminated in the approval of the first agent targeting the IFN pathway for the treatment of patients with SLE [104], highlighting that type I interferon is a viable therapeutic target that reduces disease activity in patients with SLE and other autoimmune diseases.

## Figures and Tables

**Figure 1 ijms-22-11286-f001:**
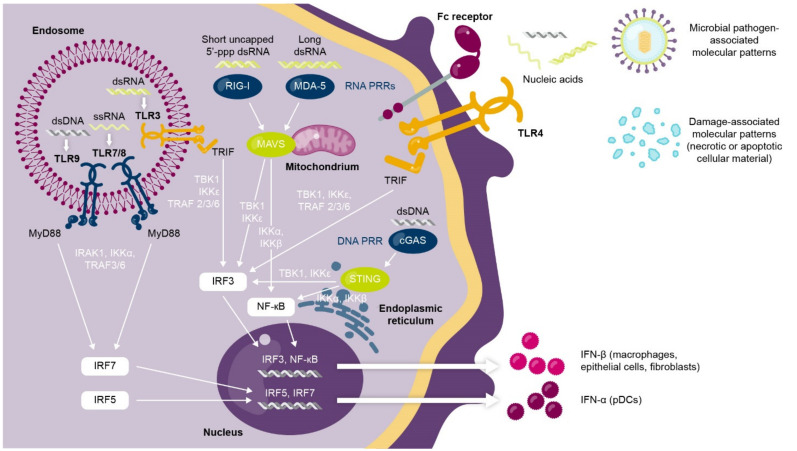
Activation of type I interferon response by viral nucleic acids and pattern recognition signaling pathways. Pattern recognition receptors located in the endosome (e.g., TLR3/7/8/9), the cell membrane (e.g., TLR4), or the cytoplasm (e.g., cGAS, RIG-I, MDA-5) detect viral nucleic acids to trigger a signaling pathway that results in type I interferon production from pDCs, macrophages, epithelial cells, and fibroblasts. cGAS, cGAMP synthase; ds, double-stranded; IFN, interferon; IKK, inhibitor of kappa B kinase; IRAK, interleukin 1 receptor-associated kinase; IRF, interferon regulatory factor; MAVS, mitochondrial antiviral-signaling protein; MDA-5, melanoma differentiation-associated protein 5; MyD88, myeloid-differentiation primary response protein 88; NF, nuclear factor; pDC, plasmacytoid dendritic cell; PRR, pattern recognition receptor; RIG-1, retinoic acid inducible gene-I; ss, single-stranded; STING, stimulator of interferon genes; TLR, Toll-like receptor; TBK1, TRAF family associated NF-κB activator (TANK)-binding kinase-1; TRAF, tumor necrosis factor receptor-associated factor; TRIF, TIR domain-containing adaptor inducing interferon-β.

**Figure 2 ijms-22-11286-f002:**
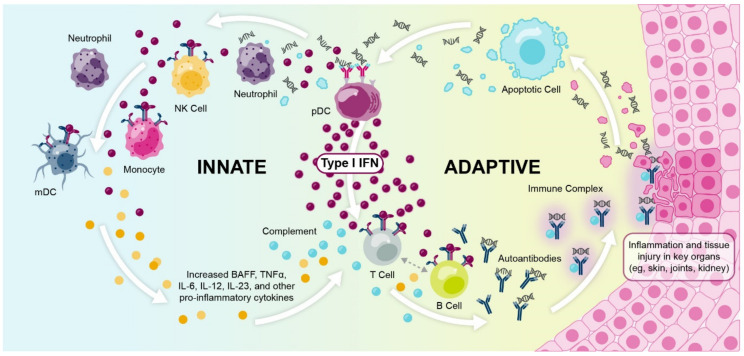
Type I interferons and immune complex formation contributes to organ damage in the cycle of SLE pathophysiology. BAFF, B-cell activating factor; IFN, interferon; IL, interleukin; mDC, myeloid dendritic cell; NK, natural killer; pDC, plasmacytoid dendritic cell; SLE, systemic lupus erythematosus; TNFα, tumor necrosis factor alpha.

**Figure 3 ijms-22-11286-f003:**
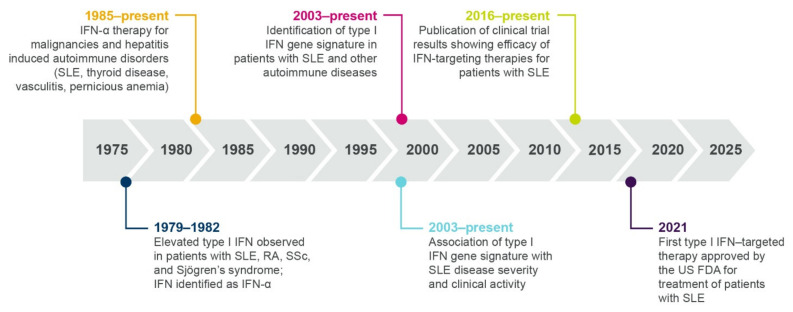
Timeline showing multiple lines of evidence that support the role of type I interferon in SLE and autoimmune pathogenesis. IFN, interferon; RA, rheumatoid arthritis; SLE, systemic lupus erythematosus; SSc, systemic sclerosis.

**Figure 4 ijms-22-11286-f004:**
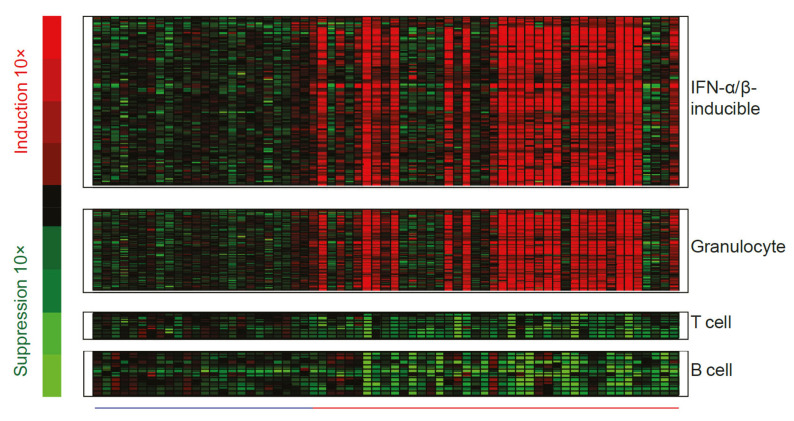
Representative heat map visualizing genes differentially expressed in whole blood from patients with SLE relative to healthy controls. The heatmap includes 110 upregulated interferon-α/β-inducible transcripts, measured in whole blood samples from patients with SLE (―; *n* = 41) and compared with whole blood samples from healthy controls (―; *n* = 24). Figure reproduced from Yao et al. Hum Genomics Proteomics 2009, 374312 [54]. IFN, interferon; SLE, systemic lupus erythematosus.

**Figure 5 ijms-22-11286-f005:**
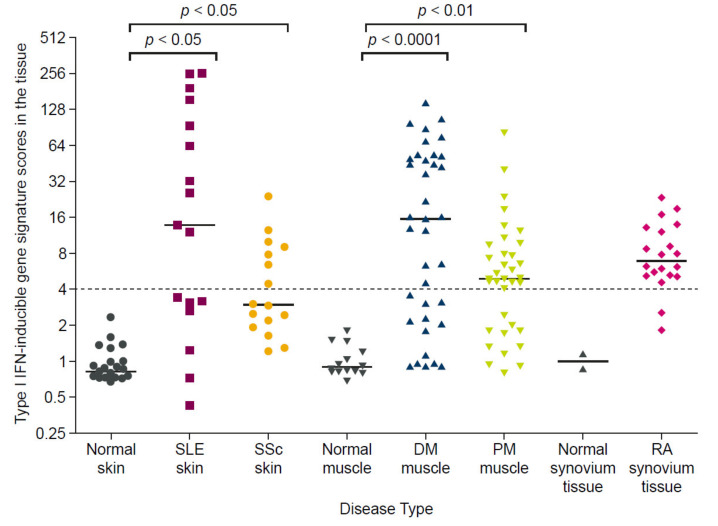
Elevated type I interferon gene signature expression in tissues of patients with SLE, SSc, DM, PM, and RA relative to healthy controls. The type I interferon gene signature score was calculated by measuring the expression of 5 type I IFN-inducible genes in disease target tissue and expressed as a fold-change relative to healthy controls. Horizontal bars represent the median values for each group and the gray dashed line indicates the threshold for signature positive or negative status. Lesional skin samples from patients with SLE (*n* = 16) and SSc (*n* = 16) had significantly greater IFNGS expression than samples from healthy controls (*n* = 25) (*p* < 0.05 for both). Muscle biopsy samples from patients with DM (*n* = 37) and PM (n=36) had significantly greater IFNGS expression than those from healthy controls (*n* = 14) (*p* < 0.0001 and *p* < 0.01, respectively). No statistical test was performed to compare the IFNGS expression in synovium tissues from patients with RA versus healthy controls, owing to the small sample size (*n* = 2 vs. *n* = 20). Figure reproduced from Higgs, B.W. et al. Patients with systemic lupus erythematosus, myositis, rheumatoid arthritis and scleroderma share activation of a common type I interferon pathway. Ann Rheum Dis 2011, 70, (11), 2029-36. 2021 [30], with permission from BMJ Publishing Group Ltd. DM, dermatomyositis; IFN, interferon; PM, polymyositis; RA, rheumatoid arthritis; SLE, systemic lupus erythematosus; SSc, systemic sclerosis.

**Figure 6 ijms-22-11286-f006:**
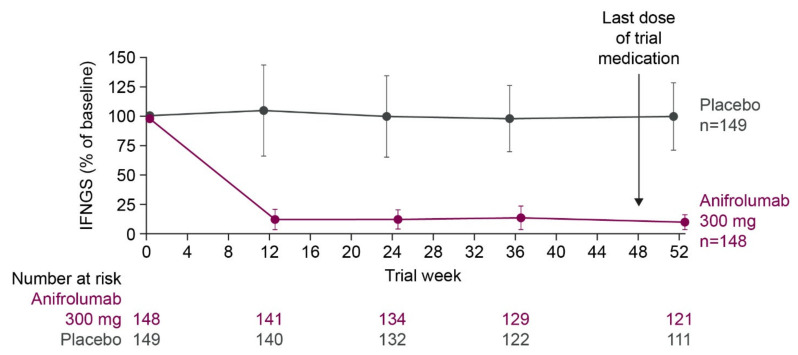
Median percentage of baseline IFNGS score throughout the treatment period in IFNGS-high patients with SLE in a phase 3 trial of anifrolumab versus placebo alongside background standard therapy. IFNGS was measured using the expression of 21 IFN-α/β inducible genes and expressed as a median percentage of baseline score. Error bars represent median absolute deviation. All patients received background therapy with oral glucocorticoids and/or immunosuppressants. Figure reproduced from Morand, E.F. et al. Trial of anifrolumab in active systemic lupus erythematosus. New Engl J Med 2020, 382, (3), 211–221 [50]. 2021 Massachusetts Medical Society. Reprinted with permission from Massachusetts Medical Society. IFN, interferon; IFNGS, interferon gene signature; SLE, systemic lupus erythematosus.

**Figure 7 ijms-22-11286-f007:**
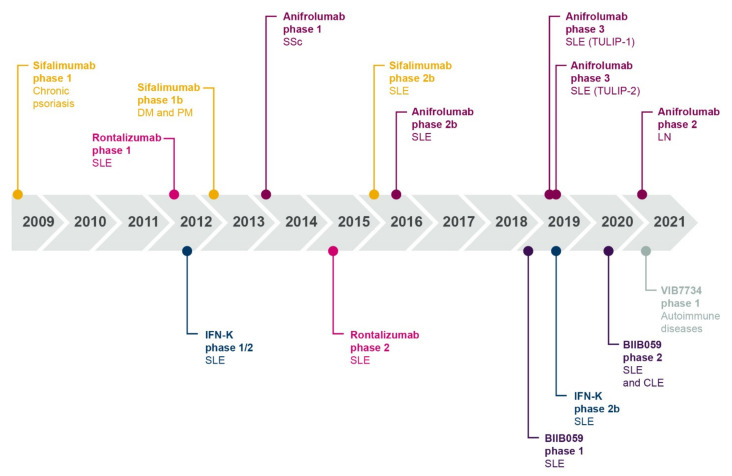
Timeline showing dates of publication of clinical trial data for interferon-targeting biologics in patients with autoimmune disease. CLE, cutaneous lupus erythematosus; DM, dermatomyositis; IFN, interferon; IFN-K, interferon kinoid; LN, lupus nephritis; PM, polymyositis; SSc, systemic sclerosis; SLE, systemic lupus erythematosus.

**Table 1 ijms-22-11286-t001:** Summary of approved and developmental therapeutics for patients with SLE and their direct/indirect effects on type I interferon signaling.

Standard Therapies				
Therapy		Impact on Type I Interferon	Treatment Recommendations	
Immunosuppressives		One recent study reported that MPA, a metabolite of MMF, inhibited production of IFN-α by pDCs from healthy donors in a dose-dependent manner, which was confirmed by transcript levels [98]; further investigation is required to see if the same effect occurs in patients with SLE	High intensity recommended for treatment of flares, followed by longer period of less intensive therapy to consolidate response and prevent relapse [92]	
Antimalarials		Multiple studies have reported inhibition of IFN-α production or IFNGS expression in patients with SLE following HCQ treatment [25,99,100]	Recommended for all patients with SLE unless contraindicated [92]	
Glucocorticoids		-	Pulses of intravenous methylprednisolone recommended for short term; oral glucocorticoids should be tapered to 7.5 mg/day and, where possible, withdrawn [92]	
**Approved Biologics**				
**Therapy**		**Impact on Type I Interferon**	**Treatment Recommendations**	
Belimumab [101], anti-BLyS mAb		Belimumab treatment reduced levels of both IFN-α protein and circulating autoantibodies in patients with SLE [101]	Indicated for the treatment of patients aged 5 years or older with active, autoantibody-positive SLE who are receiving standard therapy, and adult patients with active LN who are receiving standard therapy [102]	
Anifrolumab; anti-IFNAR1 mAb [103,104]		Blocks signaling of all type I IFNs, suppresses IFNGS [103,104]	Indicated for the treatment of adult patients with moderate to severe SLE who are receiving standard therapy [105,106]	
**Other IFN-Targeting (Direct and Indirect) Therapeutics in Clinical Development**				
**Target**	**Clinical Trial Phase**	**Impact on Type I Interferon**	**Primary Efficacy Outcome**	**Efficacy Results**
IFN-κ; therapeutic vaccine composed of inactivated IFNα2b coupled to carrier protein [107,108]	Phase 2b (discontinued) [109]	Induces production of neutralizing polyclonal anti-IFN-α-antibodies leading to normalization of the IFNGS [107,108]	Coprimary endpoints: Neutralization of IFNGS, and modified BICLA response (requiring glucocorticoid taper) at Week 36 [109]	BICLA primary endpoint not met.More patients attained LLDAS or glucocorticoid tapering with IFN-K than with placebo [109]
Sifalimumab, anti-IFN-α mAb [53]	Phase 2 (discontinued)	Neutralizes IFN-α, suppresses IFNGS in blood and skin [7]	SRI(4) response at Week 52 [53]	Primary and secondary endpoints were met, with greater proportions of patients achieving an SRI(4) response at Week 52 with sifalimumab versus placebo, accompanied by other improvements in skin disease activity and reduction in swollen and tender joint counts [53]
Rontalizumab; anti-IFN-α mAb [110]	Phase 2 (discontinued)	Neutralizes IFN-α levels and suppresses IFNGS [110]	BILAG-2004 reduction at Week 24 [111]	Primary and secondary (SRI [4] response) endpoints not met [111]
**pDC Inhibition/Depletion Strategies**				
**Target**	**Clinical Trial Phase**	**Impact on Type I Interferon**	**Primary Efficacy Outcome**	**Efficacy Results**
VIB7734, pDC-depleting anti-ILT7 mAb [112]	Phase 1 [112] (phase 2 trial recruiting [113])	Depleted levels of circulating and tissue-resident pDCs and decreased local type I IFN production [112]	Safety and tolerability [112]	Improvement from baseline in skin disease activity with VIB7734 versus placebo [112]
BIIB059, pDC-inhibitory anti-BDCA2 mAb [114,115]	Phase 2 [116]	BIIB059 inhibits IFN production from pDCs, leading to decreased IFNGS expression in blood and normalized IFN response proteins in affected skin [114,115]	Change in total active joint count from baseline to Week 24 [116]	BIIB059 treatment was associated with reduced active joint counts and higher SRI(4) response rates compared with placebo in patients with SLE [116]
Small molecule JAK inhibitors (multiple)	Various	Suppress IFNGS expression [117,118,119]	Various	Tofacitinib treatment was associated with fewer low-density granulocytes (signs of dysregulated neutrophil function) and improvements in cardiometabolic parameters and vascular function [117]

BDCA2, blood dendritic cell antigen 2; BICLA, BILAG-based Composite Lupus Assessment; BLyS, B-lymphocyte stimulator; HCQ, hydroxychloroquine; IFN, interferon; IFNAR1, interferon alpha receptor 1; IFNGS, interferon gene signature; IFN-K, interferon kinoid; ILT7, immunoglobulin-like transcript 7; JAK, janus kinase; LLDAS, lupus low disease activity state; LN, lupus nephritis; mAb, monoclonal antibody; MMF, mycophenolate mofetil; pDC, plasmacytoid dendritic cell; SLE, systemic lupus erythematosus; SRI(4), Systemic Lupus Erythematosus Responder Index ≥4.

## Data Availability

Not applicable.

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
