# Peer review of "The Pathogenesis, Molecular Mechanisms, and Therapeutic Potential of the Interferon Pathway in Systemic Lupus Erythematosus and Other Autoimmune Diseases"

_ijms, 2021, doi:10.3390/ijms222011286_

Round 1
Reviewer 1 Report
This manuscript is a very detailed and comprehensive review of the literature associated to interferons in autoimmune diseases, with an emphasis on lupus.
Some suggestions:
- Because IFNs are so much linked to autoimmune diseases in general in the manuscript, maybe the title should reflect it (e.g. “… in autoimmune diseases and SLE”).
- I would maybe mention APOL1 somewhere, as it is linked to the increased phenotype severity and plays a role in inflammatory response.
- Some recent references were missing regarding IFN responses in SLE; also regarding the relationship of molecular SLE profiles with IFN.
- Single cell paragraph could have mentioned more recent work published on SLE tissues; this could be linked to the cellular origin of IFNs, for targeting specific cell type producing IFN.
Author Response
This manuscript is a very detailed and comprehensive review of the literature associated to interferons in autoimmune diseases, with an emphasis on lupus. Some suggestions:
- Because IFNs are so much linked to autoimmune diseases in general in the manuscript, maybe the title should reflect it (eg, “… in autoimmune diseases and SLE”).
Response: We thank the reviewer for their comments. We have amended the title to “The Pathogenesis, Molecular Mechanisms, and Therapeutic Potential of the Interferon Pathway in Systemic Lupus Erythematosus and Other Autoimmune Diseases”
- I would maybe mention APOL1 somewhere, as it is linked to the increased phenotype severity and plays a role in inflammatory response.
Response: We have added wording to discuss the impact of interferon upregulation on phenotypic severity of APOL1 risk alleles (page 16 of tracked-changes manuscript): “Furthermore, interferon signaling upregulation in patients with LN may trigger nephropathy associated with apolipoprotein L1 (APOL1) risk-alleles (which are prevalent in individuals with sub-Saharan African ancestry) by upregulating APOL1 mRNA expression and increasing genotype-associated disease severity (Vajgel et al, 2020).”
- Some recent references were missing regarding IFN responses in SLE; also regarding the relationship of molecular SLE profiles with IFN.
Response: We have added a new paragraph that includes additional references on interferon responses in patients with SLE and details studies assessing interferon dysregulation within the broader context of SLE molecular profiles (pages 16–17 of tracked-changes manuscript): “Decades of profiling gene and protein expression has confirmed the molecular heterogeneity of SLE, and recent studies have attempted to contextualize interferon upregulation within broader immune dysregulation. In a transcriptomics study of pediatric patients with SLE, ‘personalized immunomonitoring’ over time revealed gene expression modules, supported by genotypes, that enabled patient stratification by disease activity. These modules included interferon response genes as well as genes categorized by plasmablast, cell cycle, neutrophil, histone, and B-cell functions (Banchereau R et al, 2016). A separate investigation of longitudinal gene expression identified multiple gene modules associated with neutrophil and lymphocyte activity, and differentially associated with type I interferon activity, that allowed adult and pediatric patients with SLE to be stratified by disease activity (Toro-Domínguez et al, 2018). Another study assessing gene co-expression across different immune cell types in patients with SLE, identified coordinated interferon responses and cross-correlation with T-helper cell markers and B-cell responses that in turn correlated with disease severity (Panwar et al, 2021). These findings are relevant because stratification by disease activity may provide rationale for clinical trial failures/successes and may help to identify subpopulations suited for targeted therapies.”
- Single cell paragraph could have mentioned more recent work published on SLE tissues; this could be linked to the cellular origin of IFNs for targeting specific cell-type producing IFN.
Response: As the cellular origin of interferon production was the focus of an earlier section (page 7 of tracked-changes manuscript), we added additional wording on the cellular origin of interferons: "pDCs play an important role in the aforementioned autoamplification loop that drives type I interferon production and downstream autoantibody generation, as antigen-presentation and type I interferon production take place simultaneously in pDCs and can recruit new pDCs to tissues (Rönnblom et al, 2001).”
The single cell paragraph has also been amended to reflect the addition of 3 new references and discussion of the potential for targeting specific IFN-producing cells (page 15 of tracked-changes manuscript): “Recently, single-cell analyses have been utilized to investigate the cellular source of the elevated type I interferon gene signature detected in serum and tissue of patients with SLE or LN (ie, the cells responding to the aberrantly produced pDC-derived type I interferon). These studies identified elevated interferon-regulated gene expression in a broad range of immune cell types (eg, B cells, natural killer cells, monocytes, T cells, conventional dendritic cells, pDCs, neutrophils, and low-density granulocytes) (Nehar-Belaid et al, 2020; Chen et al, 2021; Jin et al, 2017; Deng et al, 2021). Furthermore, single-cell RNA sequencing of renal biopsies from patients with LN identified type I interferon responses in keratinocytes and tubular cells that distinguished patients with LN from healthy controls (Der et al, 2019). The broad range of immune cells types expressing the IFNGS likely reflects the ubiquity of IFNAR1 expression across many immune cell types (Rönnblom et al, 2019; de Weerd et al, 2012). Signal blockade via the IFNAR1 thereby has the potential to affect disease biology across a range of immune cell types, including B cells, T cells, pDCs, and natural killer cells (de Weerd et al, 2012); these studies also highlight the potential for targeting specific IFN-producing cell types (such as pDCs, discussed in a later section).”
We also clarified that a study mentioned on page 20 of tracked-changes manuscript was a recent single-cell analysis: “In a separate single-cell transcriptomic analysis of renal tissue from patients with LN, patients who did not respond to standard therapy had greater expression of type I interferon response genes in tubular cells than those who achieved at least a partial response (Der et al, 2019).”
References
Banchereau R, et al. Cell 2016, 165, 551–565.
Chen Z, et al. J Cell Mol Med 2021, 25, 4684–4695.
de Weerd N, et al. Immunol Cell Biol 2012, 90, 483–491.
Deng Y, et al. EBioMedicine 2021, 70, 103477.
Der E, et al. Nat Immunol 2019, 20, 915–927.
Jin Z, et al. Lupus Sci Med 2017, 4, e000202.
Nehar-Belaid D, et al. Nat Immunol 2020, 21, 1094–1106.
Panwar B, et al. Genome Res 2021, 31, 659–676.
Rönnblom L, et al. J Exp Med 2001, 194, F59–F63.
Rönnblom L, et al. Lupus Sci Med 2019, 6, e000270.
Toro-Domínguez D, et al. Arthritis Rheumatol 2018, 70, 2025–2035.
Vajgel G, et al. J Rheumatol 2020, 47, 1209–1217.
Reviewer 2 Report
I read with interest the manuscript entitled “The Pathogenesis, Molecular Mechanisms, and Therapeutic Po-tential of the Interferon Pathway in Systemic Lupus Erythematosus” by Madhu Ramaswamy, et al” that is intended to be published in IJMS as a review.
I enjoyed reading the manuscript and did not find any inconvenience to its publication. Represents a triumph in research in the field of autoimmune diseases
Author Response
I read with interest the manuscript entitled, “The Pathogenesis, Molecular Mechanisms, and Therapeutic Potential of the Interferon Pathway in Systemic Lupus Erythematosus,” by Madhu Ramaswamy, et al. that is intended to be published in IJMS as a review.
I enjoyed reading the manuscript and did not find any inconvenience to its publication. This represents a triumph in research in the field of autoimmune diseases.
Response: Many thanks to the reviewer for their positive comments
Reviewer 3 Report
This manuscript reports a comprehensive and updated review on the pathogenetic role of Interferon in Systemic lupus erithematosus and on the potential efficacy of Interferon-blocking agents in Lupus therapy.
Author Response
This manuscript reports a comprehensive and updated review on the pathogenetic role of interferon in systemic lupus erythematosus and on the potential efficacy of interferon-blocking agents in lupus therapy.
Response: Many thanks to the reviewer for their positive comments.